# Quantification of *Enterocytozoon hepatopenaei* (EHP) in Penaeid Shrimps from Southeast Asia and Latin America Using TaqMan Probe-Based Quantitative PCR

**DOI:** 10.3390/pathogens8040233

**Published:** 2019-11-12

**Authors:** Patharapol Piamsomboon, Seong-Kyoon Choi, Bambang Hanggono, Yani Lestari Nuraini, Fatma Wati, Kathy F. J. Tang, Song Park, Dongmi Kwak, Man Hee Rhee, Jee Eun Han, Ji Hyung Kim

**Affiliations:** 1Department of Veterinary Medicine, Faculty of Veterinary Science, Chulalongkorn University, Bangkok 10330, Thailand; 2Core Protein Resources Center, DGIST, Daegu 42988, Korea; cskbest@dgist.ac.kr (S.-K.C.); cristaling@dgist.ac.kr (S.P.); 3Division of Biotechnology, DGIST, Daegu 42988, Korea; 4Fish Health and Environmental Laboratory, Brackishwater Aquaculture Development Center, Situbondo 68351, Indonesia; bambanghanggono@gmail.com (B.H.); yani_bbaps@yahoo.co.id (Y.L.N.); fatmawatiaziz06@gmail.com (F.W.); 5Yellow Sea Fisheries Research Institute, Chinese Academy of Fishery Sciences, Qingdao 266071, China; ktangnelson@gmail.com; 6College of Veterinary Medicine, Kyungpook National University, Daegu 41566, Korea; dmkwak@knu.ac.kr (D.K.); rheemh@knu.ac.kr (M.H.R.); 7Infectious Disease Research Center, Korea Research Institute of Bioscience and Biotechnology, Daejeon 34141, Korea

**Keywords:** *Enterocytozoon hepatopenaei*, qPCR, microsporidian, shrimp parasite

## Abstract

We developed a qPCR assay based on the *β*-tubulin gene sequence for the shrimp microsporidian parasite *Enterocytozoon hepatopenaei* (EHP). This assay reacted with the hepatopancreas (HP) of EHP-infected shrimps, and the highest copy numbers were found in HP and feces samples from Southeast Asian countries (10^6^–10^8^ copies mg^−1^), while HP samples from Latin America, *Artemia*, and EHP-contaminated water showed lower amounts (10^1^–10^3^ copies mg^−1^ or mL^−1^ of water). No false positive was found with the normal shrimp genome, live feeds, or other parasitic diseases. This tool will facilitate the management of EHP infection in shrimp farms.

The microsporidian *Enterocytozoon hepatopenaei* (EHP), an intracellular spore-forming parasite, is a major threat to the commercial shrimp industry. The infection is specific to the tubular epithelial cells of the hepatopancreas (HP) of shrimps. Shrimps in an advanced stage of EHP infection show signs of lethargy, reduced feed intake, soft shells, and an empty midgut [1]. This parasite was first recorded in Thailand in 2003, as an unknown microsporidian species infecting the HP of the black tiger shrimp *Penaeus monodon* [2]. It was characterized in 2009 as a new member of the genus *Enterocytozoon*, in the family Enterocytozoonidae [3]. Subsequently, it has been found in the more economically important Pacific white shrimp *P. vannamei* cultured in various Asian countries, including China, Thailand, Indonesia, Malaysia, Vietnam, and India [4,5,6,7]. Recently, EHP has also been reported in Venezuela, South America [8].

Severe EHP infections have become common in ponds, and result in substantial economic losses to shrimp producers, since they are associated with inhibited growth in cultured shrimps [1]. EHP infections have also been reported in normal shrimp reared in earthen ponds [5]. Since growth inhibition may depend upon the degree of EHP infection, a means of convenient quantitation of the amount of EHP in infected individuals is needed.

Several EHP diagnostic methods based on SSU rRNA are available, including PCR, in situ hybridization, and loop-mediated isothermal amplification assays [1,2,3,4,9]. Moreover, the qPCR assay has been developed to determine the number of EHP, which is useful for the study of disease progress and quantification of the EHP [10]. However, SSU rRNA sequences are highly conserved among microsporidia, so this may generate false positives in non-shrimp samples [11,12]. On the contrary, the *β*-tubulin gene sequence has sufficient variability for molecular diagnosis at the species level [13]. For more specific EHP quantification, we developed a qPCR assay based on the *β*-tubulin gene.

In this study, we developed a qPCR assay based on the *β*-tubulin gene sequence for EHP-specific quantification. This assay was then applied to examine shrimp and feces samples collected from various countries to quantify the level of EHP present. The method was also applicable for screening live shrimp feeds and water samples.

The primers and TaqMan probe were designed based on the *β*-tubulin gene of the EHP genome (GenBank no. KY593130), using Primer Express v. 3.0 software (Life Technologies). The DNA fragment (84-bp) was amplified using the forward primer (EHP-bTub-F2: 5′-GATTTGAGAAAATTGGCGGTTAA-3′) and the reverse primer (EHP-bTub-R2: 5′-TTCTGAACACAGAGGCGCATA-3′), while the TaqMan probe (5′-TGATTCCATTTCCACGACTGCACTTTTTC-3′) was synthesized and labeled with 6-carboxyfluorescein (FAM) on the 5′ end and *N*,*N*,*N*′,*N*′-tetramethyl-6-carboxyrhododamine (TAMRA) on the 3′ end. The qPCR assay was performed by combining 1 µL extracted DNA and a qPCR mixture containing 2.5 μl TaqMan Faster Universal PCR Master Mix (Life Technologies), 0.3 μM of each primer, and 0.1 μM TaqMan probe for a final volume of 10 μl. Amplification was performed by thermal cycling for 20 s at 95 °C followed by 40 cycles of 3 s at 95 °C and 30 s at 60 °C. Amplification and data analysis were performed using a StepOnePlus PCR system (Life Technologies).

A positive control was generated using a sequence of the *β*-tubulin gene from the EHP isolate 16-505/HP (Table 1) and cloned into the pGEM-T-Easy vector (Promega). By sequencing analysis, four clones from the isolate resulted in only one allele (100% identity), indicating the *β*-tubulin gene is a single copy gene in the EHP genome. The plasmid was designated pEHP-tubB-2 (3885-bp) and the concentration was determined by measuring OD_260nm_ (2.38 × 10^8^ copies per 1 ng of DNA). For a standard curve, the purified pEHP-tubB-2 plasmid was used, and the sensitivity was determined by preparing 10-fold dilutions. A standard curve presenting concentrations from 10^2^–10^8^ copies µL^−1^ is shown in Appendix A. The specificity of the qPCR assay was evaluated using qPCR on DNA from the shrimp parasites *Perezia* sp. microsporidium [14], *Paramoeba* sp. amoeba [15], and EHP microsporidium as a positive control [6]. The results confirm that the primers and probe from the *β*-tubulin gene sequence are specific to EHP, showing no reaction to another shrimp parasites. In addition, DNA from different shrimp species and live shrimp feed were tested to ensure that the primers did not cause false positives (Table 1).

To evaluate practical use, a total of 86 samples were collected from a population of EHP-affected *P. vannamei* (Table 2). These samples included 14 HP and three fecal samples from Thailand, 23 HP samples from Vietnam, 36 HP and one fecal sample from Indonesia, four HP samples from Venezuela, four *Artemia* biomass samples, and one water sample. For DNA extraction, feces were collected by siphoning onto a 500 μm mesh screen, and the water sample was concentrated (1:100) using a Microsep Advance Centrifugal Device (PALL Corporation). Approximately 30 mg of HP, feces, and artemia samples and 300 μL of water samples were used for DNA extraction using QIAamp tissue kit (Qiagen). After DNA extraction, the concentration of DNA was measured by a spectrophotometer (BioPhotometer, Eppendorf). The EHP copy number was highest in the HP samples from Indonesia, with a range between 8.0 × 10^3^–1.4 × 10^8^ copies mg^−1^, followed by those from Thailand (2.5 × 10^2^–9.1 × 10^7^ copies mg^−1^) and Vietnam (1.9 × 10^3^–4.8 × 10^7^ copies mg^−1^). In contrast to the SE Asian samples, HP samples from Latin America presented a relatively low EHP quantity, with copy numbers between 2.1 × 10^1^–1.3 × 10^2^ copies mg^−1^. In the feces sample, 10^6^–1.7 × 10^7^ copies mg^−1^ were found, while *Artemia* samples contained EHP DNA at 4.3 × 10^3^–2.8 × 10^5^ copies mg^−1^. Finally, EHP was detected in the water samples at 1.7 × 10^3^ copies per mL^−1^.

All samples tested in this study came from shrimps exhibiting slow growth in farms from SE Asia and Latin America, and from contaminated *Artemia*. This indicates good applicability of the qPCR assay regardless of the source of EHP and their host species. For the *Artemia* samples, substantial amounts of EHP (as DNA copies) were detected. Live shrimp feeds, including polychaetes, squids, and *Artemia* biomass, have been suspected to be carriers for EHP, so EHP quantification by qPCR in live shrimp feeds can be used as a tool to aid in disease prevention in broodstock populations. Using qPCR, we also confirmed that feces contained high numbers of EHP copies, indicating that this may act as an important source of infection in ponds. In penaeid shrimps, EHP multiplies rapidly in the HP tubule epithelium cells, and EHP spores are released into the digestive system, causing white feces in ponds [6].

In conclusion, EHP infection is notorious in farmed shrimp ponds, and such EHP infections are likely to continue to be prevalent. The situation requires careful monitoring of EHP levels in shrimps and their feed using a highly specific diagnostic method. In addition, quantification of EHP in live feeds, pond environments, or feces becomes more important for EHP management. This study describes a new quantification method for EHP based on the *β*-tubulin gene, applicable in EHP management in shrimp farming.

## Figures and Tables

**Table 1 pathogens-08-00233-t001:** Samples used for testing the specificity of the qPCR assay.

Sample	Number of Samples	Number of Positive Samples
Pathogen-free *P. vannamei*	3	0
*P. monodon*	3	0
*P. indicus*	3	0
*P. stylirostris*	2	0
*P. aztecus*	2	0
*Macrobrachium rosenbergii*	2	0
Crabs (unknown species)	3	0
Polychaetes	4	0
*Artemia*	3	0
Plankton	1	0
*P. vannamei* with *Paramoeba* sp. infection	1	0
*P. monodon* with *Perezia* sp. infection	1	0
*P. vannamei* (HP) with EHP infection	77	77
Feces with EHP infection	4	4
*Artemia* with EHP infection	4	4
Water with EHP infection	1	1

**Table 2 pathogens-08-00233-t002:** Sample types tested and quantification of *Enterocytozoon hepatopenaei* (EHP) by qPCR.

Case No.	Origin	Year	No. of Samples	Sample Types	Copy Number mg^−1^
14-435/1	Thailand	2014	1	Feces	1.0 × 10^6^
15-220	Thailand	2015	2	Feces	2.5 × 10^3^, 1.2 × 10^6^
16-505/HP	Thailand	2016	14	HP	2.5 × 10^2^, 4.4 × 10^2^, 1.2 × 10^3^, 1.2 × 10^3^, 3.4 × 10^3^, 4.1 × 10^3^, 5.4 × 10^4^, 5.2 × 10^5^, 4.8 × 10^6^, 7.9 × 10^6^, 9.9 × 10^6^, 2.2 × 10^7^, 3.4 × 10^7^, 9.1 × 10^7^
14-480	Vietnam	2014	23	HP	1.9 × 10^3^, 3.3 × 10^4^, 1.0 × 10^5^, 1.6 × 10^5^, 2.8 × 10^5^, 2.9 × 10^5^, 3.1 × 10^5^, 3.4 × 10^5^, 3.4 × 10^5^, 6.6 × 10^5^, 1.4 × 10^6^, 1.8 × 10^6^, 2.1 × 10^6^, 2.3 × 10^6^, 2.4 × 10^6^, 3.0 × 10^6^, 3.7 × 10^6^, 4.0 × 10^6^, 4.1 × 10^6^, 4.4 × 10^6^, 6.1 × 10^6^, 1.7 × 10^7^, 4.8 × 10^7^
16-273/F	Indonesia	2016	1	Feces	6.9 × 10^7^
16-273/A	Indonesia	2016	1	HP	3.0 × 10^6^
16-273/C	Indonesia	2016	10	HP	1.0 × 10^6^, 1.6 × 10^6^, 3.9 × 10^6^, 4.4 × 10^6^, 5.3 × 10^6^, 5.5 × 10^6^, 2.6 × 10^7^, 4.3 × 10^7^, 4.5 × 10^7^, 7.9 × 10^7^
16-597	Indonesia	2016	5	HP	8.0 × 10^3^, 6.9 × 10^5^, 2.5 × 10^6^, 4.3 × 10^6^, 9.1 × 10^6^
16-663/A	Indonesia	2016	20	HP	5.0 × 10^4^, 2.0 × 10^5^, 4.0 × 10^5^, 1.6 × 10^7^, 1.8 × 10^7^, 2.4 × 10^7^, 2.5 × 10^7^, 2.6 × 10^7^, 2.7 × 10^7^, 3.2 × 10^7^, 3.2 × 10^7^, 4.3 × 10^7^, 4.5 × 10^7^, 5.1 × 10^7^, 5.2 × 10^7^, 5.5 × 10^7^, 9.0 × 10^7^, 9.6 × 10^7^, 1.0 × 10^8^, 1.4 × 10^8^
16-681	Venezuela	2016	4	HP	2.1 × 10^1^, 4.5 × 10^1^, 6.2 × 10^1^, 1.3 × 10^2^
14-428	Commercial	2014	1	Artemia	4.4 × 10^3^
14-350	Commercial	2015	1	Artemia	4.3 × 10^3^
15/025	Commercial	2015	2	Artemia	1.2 × 10^5^, 2.8 × 10^5^
14-435/2	Laboratory	2017	1	Water	1.7 × 10^3^ (Copy no. mL^−1^)

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
