# Peer review of "Quantification of Enterocytozoon hepatopenaei (EHP) in Penaeid Shrimps from Southeast Asia and Latin America Using TaqMan Probe-Based Quantitative PCR"

_pathogens, 2019, doi:10.3390/pathogens8040233_

Round 1

Reviewer 1 Report

The Brief Report entitled "Quantification of Enterocytozoon hepatopenaei (EHP) in penaeid shrimps from Southeast Asia and Latin America using TaqMan probe-based quantitative PCR" represents a further step in the pathology of breeding peneids bringing further help in the diagnosis of this Microsporidiosis.

The authors should insert the size of the gene fragment of the b-tubulin amplified and further verify the amplification conditions.

It would be appropriate to also enter the faeces sampling method to complete the descriptive picture. In lines 55-56 the numbers of samples from Venezuela, Artemia and faeces should be indicated in Roman numbers, as for the samples described above (lines 54-55).

Author Response

Response letter

Response to Reviewer 1

Thank you very much for reviewing our manuscript. The constructive comments and suggestions are most appreciated. Please find below a point-by-point response to each of the comments. We believe we have addressed all questions raised by the reviewer of the manuscript.

- Comments

: The Brief Report entitled "Quantification of Enterocytozoon hepatopenaei (EHP) in penaeid shrimps from Southeast Asia and Latin America using TaqMan probe-based quantitative PCR" represents a further step in the pathology of breeding peneids bringing further help in the diagnosis of this Microsporidiosis.

The authors should insert the size of the gene fragment of the b-tubulin amplified and further verify the amplification conditions. Thank you for the valuable comments. As suggested by the reviewer, the size of the amplified gene fragment (84-bp) was added in the text as the following “The DNA fragment (84-bp) was amplified using the forward primer (EHP-bTub-F2: 5′-GATTTGAGAAAATTGGCGGTTAA-3′) and the reverse primer (EHP-bTub-R2: 5′-TTCTGAACACAGAGGCGCATA-3′).” (Line 43). In addition, amplification conditions was described as the following “Amplification was performed by thermal cycling for 20 s at 95 °C followed by 40 cycles of 3 s at 95 °C and 30 s at 60 °C. Amplification and data analysis were performed using a StepOnePlus PCR system (Life Technologies)”. This condition is available for the TaqMan Faster Universal PCR Master Mix (Life Technologies). (Line 48)

It would be appropriate to also enter the faeces sampling method to complete the descriptive picture. In lines 55-56 the numbers of samples from Venezuela, Artemia and faeces should be indicated in Roman numbers, as for the samples described above (lines 54-55). Thank you for the valuable comments. As suggested by the reviewer, sampling method for feces were added in the manuscript as the following “For DNA extraction, feces were collected by siphoning onto a 500 μm mesh screen, and water sample was concentrated (1:100) using a Microsep Advance Centrifugal Device (PALL Corporation). Approximately 30 mg of HP, feces, and artemia samples and 300μL of water samples were used for DNA extraction using QIAamp Tissue Kit (Qiagen).” (Line 63-66). This was also commented by the reviewer #2. In addition, numbers of samples were described in Roman numbers as the following “These samples included 14 HP and 3 fecal samples from Thailand, 23 HP samples from Vietnam, 36 HP and 1 fecal sample from Indonesia, 4 HP samples from Venezuela, 4 Artemia biomass samples, and 1 water sample.” (Line 62)

Reviewer 2 Report

The manuscript by Piamsomboon et al. reports the development of real-time PCR quantification method of a shrimp parasite, Enterocytozoon hepatopenaei (EHP). Given the economic importance of shrimp production in Asia and the high prevalence of EHP, its PCR quantification is of high practical significance.

My major concern is the novelty of the methodology. The authors missed many similar studies such as Liu et al. (2018) J Invertebr Pathol 151, 191-196, and Qiu et al. (2018) Invertebr Pathol 154, 95-101, and many else. These studies used very similar approaches (TaqMan but for other genes) to detect EHP, while the authors used a new gene, beta tubulin. The tubulin gene may have some advantages over other genes as the authors stated in this paper and Ref. 14, but previous studies should be cited with appropriate discussions. The paragraph starting at L72 should be placed in the Introduction part to clarify what has been already achieved in this field. The advantage of beta tubulin over other genes should be described in the Discussion part using the results of this study.

L36: I am curious whether the beta tubulin gene is a single copy gene in the EHP genome. The specificity of this experiment should be high enough since the authors used TaqMan probes, but this information can further validate their conclusion.

L40: DNA extraction is a critical part of this study. Please provide details of the amount of samples used, DNA extraction methods, and how DNA concentrations were determined.

Table 1: Please clarify how the authors determined positive and negative results. Were there any threshold values? If not, providing representative amplification plots would be helpful.

Author Response

Response letter

Response to Reviewer 2

Thank you very much for reviewing our manuscript. The constructive comments and suggestions are most appreciated. Please find below a point-by-point response to each of the comments. We believe we have addressed all questions raised by the reviewer of the manuscript.

- Comment

: The manuscript by Piamsomboon et al. reports the development of real-time PCR quantification method of a shrimp parasite, Enterocytozoon hepatopenaei (EHP). Given the economic importance of shrimp production in Asia and the high prevalence of EHP, its PCR quantification is of high practical significance.

My major concern is the novelty of the methodology. The authors missed many similar studies such as Liu et al. (2018) J Invertebr Pathol 151, 191-196, and Qiu et al. (2018) Invertebr Pathol 154, 95-101, and many else. These studies used very similar approaches (TaqMan but for other genes) to detect EHP, while the authors used a new gene, beta tubulin. The tubulin gene may have some advantages over other genes as the authors stated in this paper and Ref. 14, but previous studies should be cited with appropriate discussions. The paragraph starting at L72 should be placed in the Introduction part to clarify what has been already achieved in this field. The advantage of beta tubulin over other genes should be described in the Discussion part using the results of this study. Thank you for the valuable comments. As suggested by the reviewer, the paragraph (starting at L72) was moved to the Introduction part (Line 33), and the reference (Liu et al., 2018) was added in the text to clarify qPCR method for this pathogen (EHP) has been already achieved in this field. (Line 35) In addition, the advantage of beta tubulin over other genes was described. (Line 36)

L36: I am curious whether the beta tubulin gene is a single copy gene in the EHP genome. The specificity of this experiment should be high enough since the authors used TaqMan probes, but this information can further validate their conclusion. Thank you for the valuable comments. To support that the beta tubulin gene is a single copy gene in the EHP genome, we have added the sequencing result of the clones as the following, “By sequencing analysis, 4 clones from the isolate (16-505/HP) resulted in only one allele (100% identity), indicating the β-tubulin gene is a single copy gene in the EHP genome.” (Line 51). In addition, results of the specificity test was described as the following “Specificity of the qPCR assay was evaluated using qPCR on DNA from the shrimp parasites Perezia sp. microsporidium [14], Paramoeba sp. amoeba [15], and EHP microsporidium as a positive control [6]. The results confirm that the primers and probe from the β-tubulin gene sequence is specific to EHP, showing no reaction to another shrimp parasites.” to confirm that there was no cross reaction for other shrimp parasites. (Line 55)

L40: DNA extraction is a critical part of this study. Please provide details of the amount of samples used, DNA extraction methods, and how DNA concentrations were determined. Thank you for the valuable comments. As suggested by the reviewer, we have added details of the amount of samples used, DNA extraction methods, and how DNA concentrations were determined as the following “For DNA extraction, feces were collected by siphoning onto a 500 μm mesh screen, and water sample was concentrated (1:100) using a Microsep Advance Centrifugal Device (PALL Corporation). Approximately 30 mg of HP, feces, and artemia samples and 300μL of water samples were used for DNA extraction using QIAamp Tissue Kit (Qiagen). After DNA extraction, concentration of DNA were measured by a spectrophotometer (BioPhotometer, Eppendorf). (Line 63-66) This was also commented by the reviewer #1.

Table 1: Please clarify how the authors determined positive and negative results. Were there any threshold values? If not, providing representative amplification plots would be helpful. Thank you for the valuable comments. As suggested by the reviewer, we have added representative amplification plots in supplementary data. (Line 93)
